# 7091-Roads and landslides in Nepal: How development affects environmental risk

McAdoo, B.G. (1)., Quak, M. (1) Gnyawali, K. R. (2), Adhikari, B. R. (3), Devkota, S. (3), Rajbhandari, P. (4), Sudmeier-Rieux, K. (5)

1.  Yale-NUS College, Singapore
2.  Natural Hazards Section, Himalayan Risk Research Institute (HRI), Nepal
3.  Department of Civil Engineering, Institute of Engineering, Tribhuvan University, Nepal
4.  Independent, Kathmandu, Nepal
5.  University of Lausanne, Faculty of Geosciences and Environment, Institute of Earth Science, Switzerland

**Abstract.** The number of deaths from landslides in Nepal has been increasing dramatically due to a complex combination of earthquakes, climate change, and an explosion of informal road construction that destabilises slopes during the rainy season. This trend will likely rise as development continues, especially as China's Belt and Road Initiative seeks to construct three major trunk roads through the Nepali Himalaya that adjacent communities will seek to tie in to with poorly-constructed roads. To determine the effect of these informal roads on generating landslides, we compare the distance between roads and landslides triggered by the 2015 Gorkha earthquake with those triggered by monsoon rainfalls, as well as a set of randomly located landslides to determine if the spatial correlation is strong enough to further imply causation. If roads are indeed causing landslides, we should see a clustering of rainfall-triggered landslides closer to the roads that accumulate and focus the water that facilitates failure. We find that in addition to a concentration of landslides in landscapes with more developed, agriculturally viable soils, that the rainfall-triggered landslides are more than twice as likely to occur within 100 m of a road than the landslides generated by the earthquake. The oversteepened slopes, poor water drainage and debris management provide the necessary conditions for failure during heavy monsoonal rains. Based on these findings, geoscientists, planners and policymakers must consider how road development affects the physical (and ecological), socio-political and economic factors that increases risk in exposed communities, alongside ecologically and financially sustainable solutions such as green roads.

## 1. Introduction

On 29 and 30 July 2015, during the first monsoon season after the Mw=7.8 Gorkha earthquake, a dramatic cloudburst triggered landslides that killed 29 people in Nepal's Western Region (**BBC, 2015**). These deadly landslides and many others like them are not solely the result of intensified rainfall associated with climate change (**Bharti et al., 2016)**, but a complex intersection of socio-economic factors with a highly-altered physical landscape where informal, non-engineered roads regularly fail during the annual monsoon season (**Petley et al., 2007; Froude and Petley, 2018**). This problem will become more acute as China's Belt and Road Initiative (BRI) aims to expand trade into Nepal, India and beyond via a series of trans-Himalayan corridors which traverse some of the world's most geomorphically-complex terrain

(**Bhushal, 2017**).  This expanded transportation network will have unintended effects on the
surrounding landscapes as villages seek to link to these highways with informal roads
constructed and maintained with severely limited resources, putting them more at risk of
landsliding.

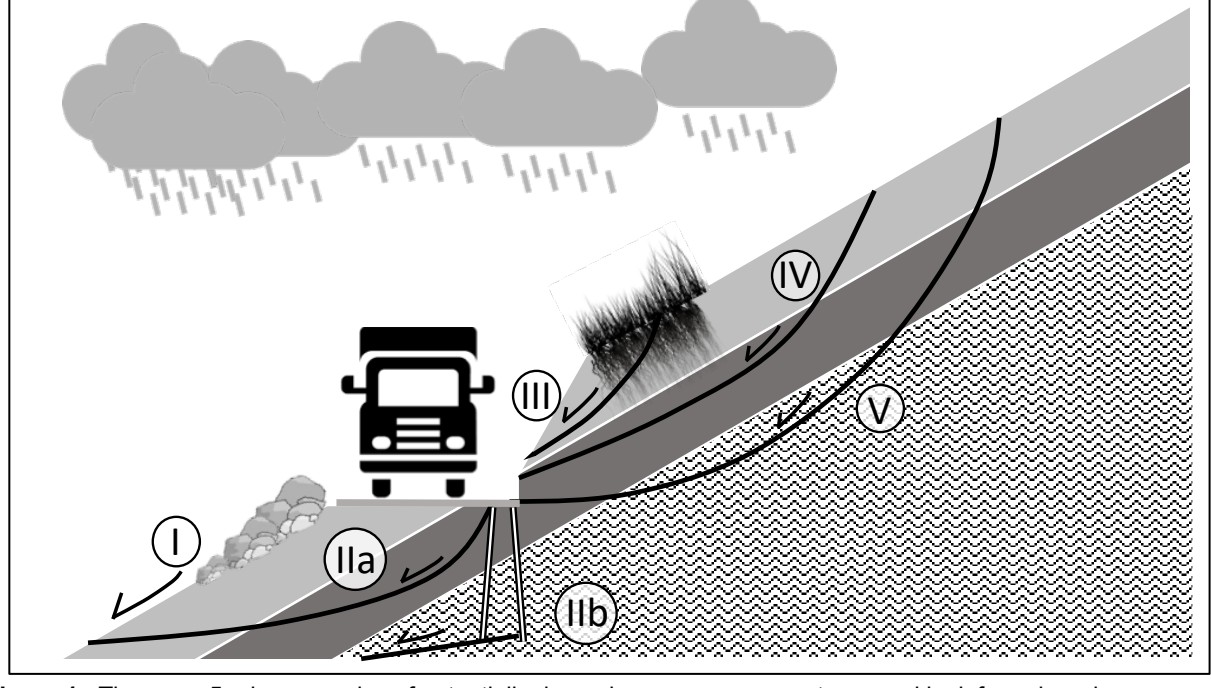


**Figure 1.**  There are 5 primary modes of potentially damaging mass movements caused by informal road
construction in Nepal- I) debris flows from excavated material stored on the downslope side of the road; II) Deeper
seated landslides that are accommodated by poor road drainage as water seepage can aid failures that include
regolith (IIa), and freeze-thaw in joints that can result in bedrock failures (IIb) ; III) Shallow failures close to the road
caused by oversteepened road cuts that may be mitigated by planting; IV) Shallow landslides caused by
oversteepening that include potentially stabilising roots from vegetation; V) Deeper seated failures triggered by
oversteepening by road cuts that may include bedrock.

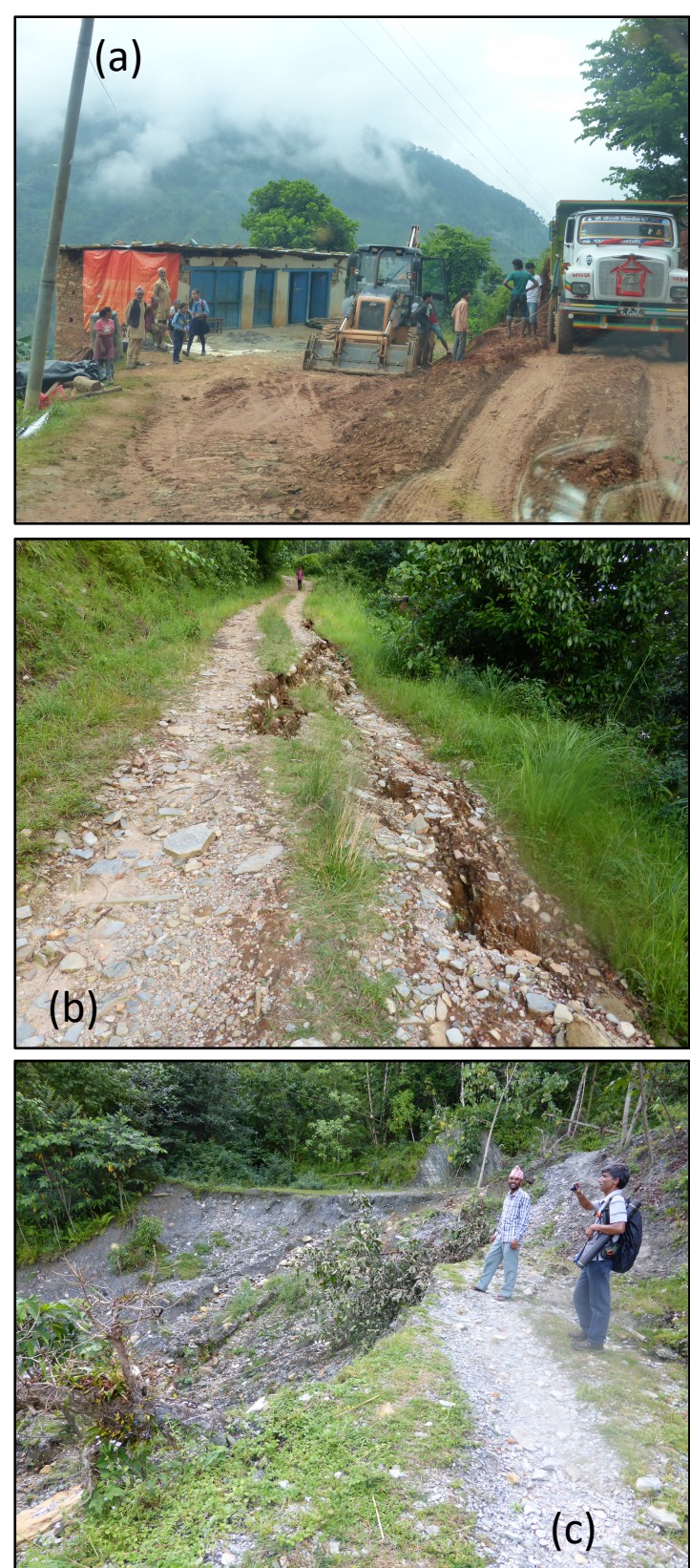

**Figure 2.** Informal, rural roads in Sindhupalchok District, Nepal. **(a)** Earth-moving equipment is hired by villagers to expand footpaths into roads that bring goods and services to isolated locations. In **(b)** and **(c),** landslides are

triggered on these informally engineered rural roads.  Poor drainage and lack of slope stabilizing measures facilitate
failures during heavy monsoonal rains, damaging land, structures, and roads, and endangering human lives and
livelihoods. Images by the authors.

The problem of roads and associated landslides has been a long recognised yet understudied
phenomenon. **Laban (1979)** provided an early quantification of the effects of human
development on the distribution of landslides in Nepal, concluding that in the nascent days of
Nepal's vehicular road development, only 5% of observed landslides were associated with
roads. While road density data is not available from this time, the density more than tripled from
13.7 km/km$^2$ in 1998 to 49.6 km/km$^2$ in 2016 (**DoR, 2002; DoR, 2017**).  **Petley et al. (2007)**
show that number of landslide fatalities in Nepal increased dramatically between 1978-2005 and
expresses concern over poorly constructed roads.  Despite this evidence of increasing losses,
there have been a limited number of studies of roads and landslides in Nepal **(Laban, 1979;**
**Bhandary et al., 2013**), and while the BRI indeed portends increases economic opportunity, it
will also bring with it an expansion of this risky road network.

Many villages in the Middle Hills region of rural Nepal are connected by simple footpaths that
limit economic and social opportunity.  As the nation continues developing, communities expand
these pathways (funded in part by remittances sent from overseas) into vehicular roads for
better access to markets, educational opportunities, and healthcare.  The resulting informal
roads often create landslides by undercutting slopes, providing pathways for water to seep into
potential slide planes, and producing debris that is easily mobilised during heavy rainfall (e.g.
**Sidle et al., 2006**; **Fig. 1**). Access to heavy machinery (**Fig. 2a**) accelerates the pace of road
construction, and the subsequent triggered landslides (**Figs. 2b and 2c**) disrupt the
transportation networks that bring much needed goods and services to and from rural
communities, damage agricultural lands in regions where subsistence farming is the norm, and
cause tens of deaths every year (**DesInventar[1], Nepal Profile, 2016**), all counteracting the
sought-after developmental gains.

To better understand the link between the development that will follow BRI-related development
and the changes in the risk landscape, we examine the relationship between roads and
landslides in the Sindhupalchok district of Central Nepal (**Fig. 3**).  The 2015 Gorkha earthquake
heavily impacted Sindhupalchok, where over 95% of the houses were severely damaged and
where over a third of the deaths occurred (**ReliefWeb, 2017**).  The earthquake also generated
thousands of co-seismic landslides in this district (**Gnyawali and Adhikari, 2017; Fig. 3a**),
many of which intersect rural roads.  By comparing the spatial distribution of slope failures
present before and those generated during the Gorkha earthquake with a randomly-distributed
suite of landslides, we present compelling evidence that landslides caused by informal roads
are a dangerous and often overlooked geomorphic agent that compromise the development
trajectory in villages that sought to gain from the road construction. Based on these results, we
show that this mode of failure should be carefully considered in studies of landslide distribution

---

[1] The mortality statistics in the DesInventar database are likely a minimum, as much of their data comes
from media reports that originate in more accessible areas.

and development planning, especially as the BRI extends the road network through the
Himalaya.
**2. Methods**
To help determine the significance of roads in the generation of landslides, we compare the
spatial and area distribution of landslides present before the Gorkha earthquake with those
triggered by the earthquake itself.  Implicit in this comparison is that the majority of landslides
present before the earthquake were generated by monsoonal rains- **Petley et al. (2007**) show
that 90% of fatal landslides occur during the rainy season (landslides that occur without fatalities
likely go unreported, therefore it is possible that there are non-fatal landslides that occur
throughout the year).  **Gnyawali and Adhikari (2017) and Roback et al. (2018)** show that the
primary controls on the distribution of the earthquake-generated landslides are geomorphology,
degree of bedrock weathering and proximity to the earthquake rupture zone, and do not
consider the effects of human alteration of the landscape.  If there is a strong spatial correlation
between the roads and either set of landslides, we can begin to better understand how
important these roads are in altering both the physical and social landscapes.
There were on the order of 20,000 landslides generated by the Gorkha earthquake (**Gnyawali**
**and Adhikari, 2017; Roback et al., 2018; Martha et al., 2016**), of which we analysed 8,238 in
Sindhupalchok district alongside a total of 252 slides visible from satellite data in the months
before the earthquake.  The pre- and post-earthquake landslide inventories we used were
created by manually digitizing the bare earth-landslide scars and deposits where visible in
Google Earth from high resolution satellite images (sub-metre), at an eye altitude of 500 meters,
corresponding to a minimum detected landslide area being around 20 square meters (**Gnyawali**
**and Adhikari, 2017**). The post-earthquake landslide inventory consists of scars and deposits
observed in the image between April 25 (main-shock day) to May 25, 2015, during the dry
season before the monsoon rains in June. The area and spatial distributions are similar to other
catalogues of the same event (**Roback et al., 2018; Martha et al., 2016; Fig. 4**) where the
primary controls are related to proximity to earthquake rupture zone and peak ground
acceleration, as well as the physical characteristics of the topography including aspect, slope,
curvature and bedrock geology.  The pre-earthquake landslide inventory consists of failures
identified in the area before the earthquake in images between October 2014 and February
2015- these include slides generated during the 2014 monsoon season as well as older slides
not yet covered by vegetation (**Malamud et al., 2004**).  We ground truthed the location and
mode of failure of many of the slides visible from the Arniko Highway- the vast majority involve
the regolith with very few deep-seated bedrock failures.
To better isolate the relationship between landslides and the roads, we limited our analysis to
the areas in Sindhupalchok district to the agricultural regions with higher road density.  The
majority of landslides (7,230 or 85% of the combined pre- and post-earthquake inventories
yielding a landslide density of 6.2 slides per km$^2$ compared to 0.5 slides per km$^2$ in the less
productive, higher elevation soils) occur in two soil types- the better developed, agriculturally
productive eutric regosols (RGe), and the less-productive humic cambisols (CMu) that occur in
higher, more arid zones (**Dijkshoorn and Huting, 2009**; **Fig. 3a**). Of the 7,091 earthquake-
triggered landslides in these two soil types, only 2,687, or 38% are in CMu (which covers 629
km$^2$ in this district with a landslide density of 4.3 slides/km$^2$), and 35 of 139 (25%, and 0.06
slides/km$^2$) pre-earthquake landslides occur in this soil type. The remaining 104 monsoon-
triggered landslides are in an area with more agricultural development in the RGe unit (530 km$^2$
in this district with a density of 0.2 slides/km$^2$), and hence more exposed communities and
roads.

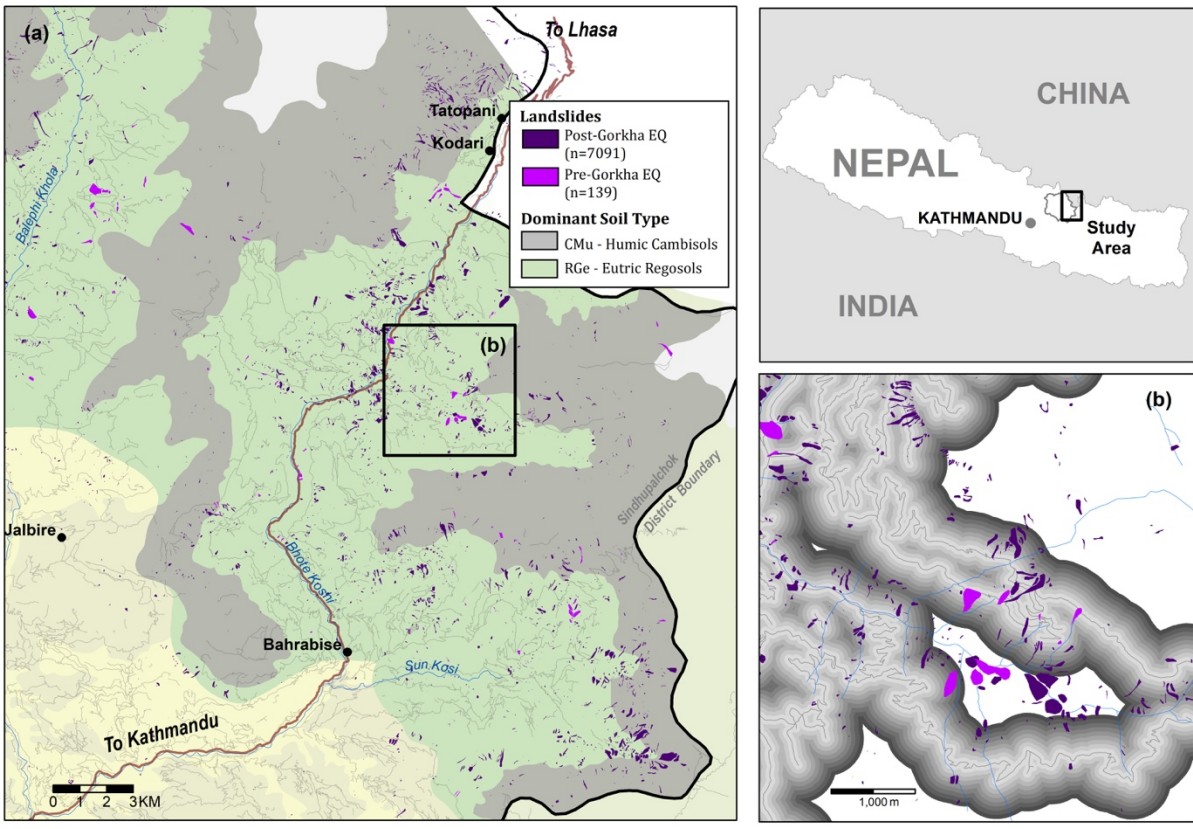

**Figure 3.** Roads and landslides in Sindhupalchok district, Nepal.  (a) The Arniko Highway that runs between
Kathmandu and Kodari at the Chinese border was heavily impacted by the 2015 Gorkha earthquake, and a dense
network of informal, rural roads grows out of this main trunk road (**OpenStreetMap Contributors, 2017**).  The red
polygons mark the location of landslides generated during the earthquake, and the blue polygons were the landslides
that were present before the earthquake (2014).  Most landslides correspond with the RGe (eutric regosols) soil type
as mapped by **Dijkshoorn and Huting (2009)**, however there is a higher percentage of earthquake-generated
failures in the humic cambisols (CMu) soils.  (b) We place buffers at 50 m intervals along the roads in the study area
that can support a vehicle to determine the distribution of landslides that correlate spatially with the roads.

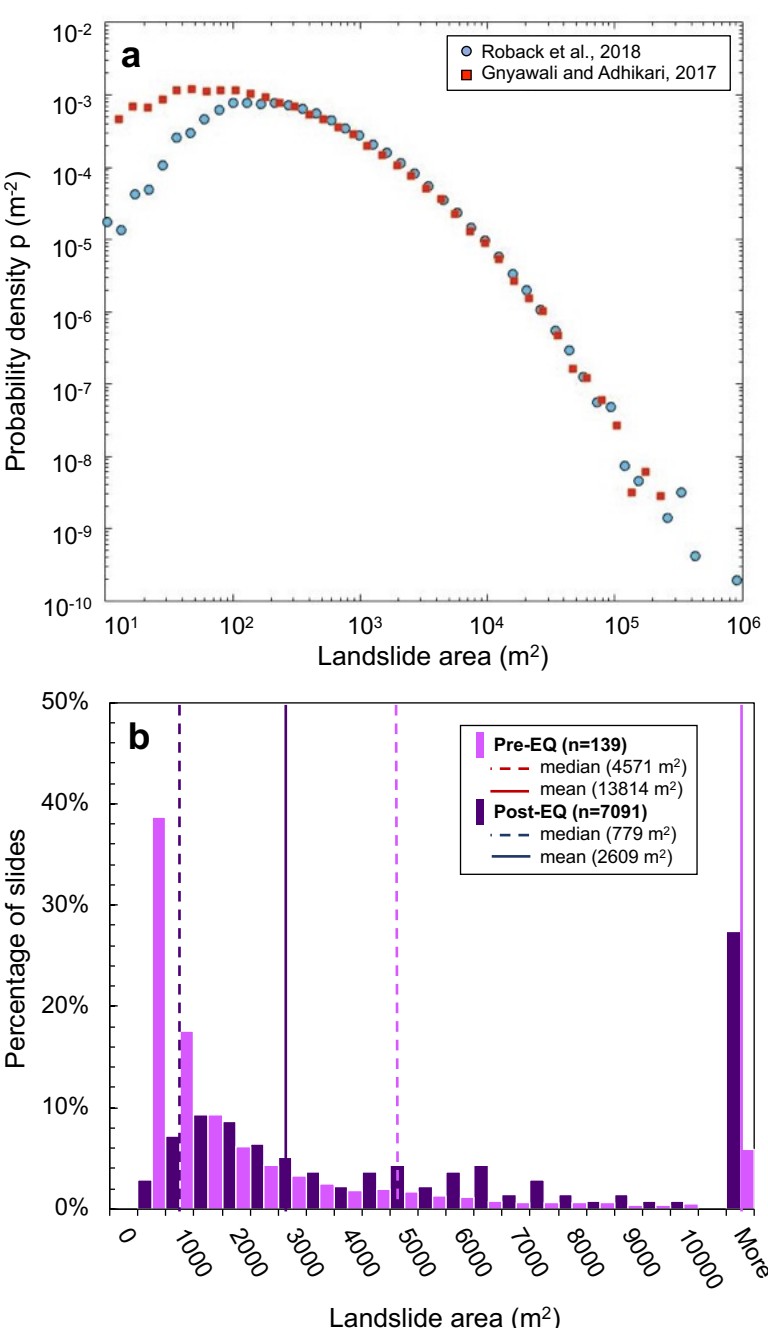

**Figure 4a and b.** (a) Probability density-area statistics of the Gorkha earthquake triggered landslide inventory used in this study compared to the inventory generated by **Roback et al. (2018)**. The two curves diverge at slides with areas less than around 200 m² suggesting that the **Gnyawali and Adhikari (2017)** data selected more smaller slides. (b) Histograms of normalised areas of landslides present before and after the 2015 Gorkha earthquake. The higher mean and median values for the monsoon-generated landslides as compared to the earthquake-generated landslides may likely reflect missed smaller, older landslides that were covered by vegetation (**Malamud et al., 2004**).

As the earthquake occurred near the end of the dry season, we expect the failures to be less affected by the presence of water, and slide location would be less influenced by features such

as roads that concentrate water.  Conversely, if as we expect there is a higher proportion of pre-
earthquake landslides near roads, it is likely that the oversteepening and poor drainage of
informal roads is indeed adding to the hazard.
To better assess the causal relationship that has been documented by many studies (e.g.
**Petley et al., 2007**; **Sidle and Ziegler, 2012; Froude and Petley, 2018**), we use a Geographic
Information System (GIS) to measure the proximity of pre- and post-earthquake slides to the
roads.  Using the existing road network (**OpenStreetMap Contributors, 2017**), we filtered out
the smallest trails and footpaths, leaving only tracks that had been improved and could likely
support a vehicle (assessment based on field observations).  We then generated nine, 50 m
buffers perpendicular to these roads (total of 450 m on each side) and tabulated the number of
landslides (scar and/or deposits) that intersected a buffer at the particular distance from the
road (Fig. 3b).
In addition, we generated 20 sets of randomised landslides (10 pre-earthquake, 10 post-
earthquake) based on the distribution landslide areas to better determine if there is a spatial
relationship of roads and failures.  For both the measured pre- and post-earthquake slides, we
plotted the cumulative log-normal area distribution, then fit a power-law curve that we used to
generate the random slide set.  For the pre-earthquake slides (n=139), the areas ($A_{pre-EQ}$) were
calculated using sets of random numbers (x)
$$A_{pre-EQ} = 0.35x^{0.097}; S=0.05 \text{ m}^2$$
and for the slides generated by the earthquake(n=7092), the areas are ($A_{post-EQ}$)
$$A_{post-EQ} = 0.44x^{0.089}; S=0.04 \text{ m}^2.$$
For each of the 20 sets of randomly generated slides, we placed them randomly within the CMu
and RGe soil types in Sindhupalchok district measuring the distances from the roads in each of
10 separate runs.  While these data lack the complex shapes of the actual landslides (they are
modelled as circular), we believe they represent a reasonable approximation of a random
distribution of failures across the landscape.
**3. Results**
Observations from the field and numerous previous studies suggest a strong spatial correlation
between roads and landslides (e.g. **Laban, 1979; Sidle et al., 2006; Petley et al., 2007;**
**Froude and Petley, 2018**), and others on how landslides affect roads (e.g. **Irigaray et al.,**
**2000**) however there have been few studies that seek to quantify the relationship with the aim of
moving past correlation to causation.  Using satellite data, we find that the majority of landslides
in Sindhupalchok district occur in the soil types that support agriculture (the eutric regosols) and
to a lesser extent, the humic cambisols) and hence have more roads.  Amongst the landslides
that were present before the 2015 earthquake, we observe a strong signal that demonstrates
the genetic relationship between agrarian development, roads, and landslides.
Although the number of monsoon-triggered landslides is small by comparison with the
earthquake-generated inventory- the total area of landslides is 1.9 km$^2$ (1.2 km$^2$ in RGe and 0.7
km$^2$ in CMu) whereas the earthquake-triggered slides cover 18.4 km$^2$ (9.8 km$^2$ in RGe and 8.6
km$^2$ in CMu).  However, it is possible that many of the smaller rainfall-induced slides may been
covered by vegetation (**Malamud et al., 2004**).  In the soil types that support agriculture, 45%
(63) of the 139 pre-earthquake landslides occur within 100 m of a road, whereas only 21%
(1,490) of the 7,091 landslides generated by the earthquake are within 100 m of a road.  Of the
randomly-generated landslides between 21% (of the post-earthquakes slide area distribution)
and 26% (of the pre-earthquake slides) of the failures are within 100 m of a road, closely
matching the spatial distribution of the earthquake landslides (**Fig. 5**).   Stated differently, there
are twice as many monsoon-generated landslides near roads than earthquake-generated
landslides, and twice as many than in a randomly located suite of slides with the same area
distribution.

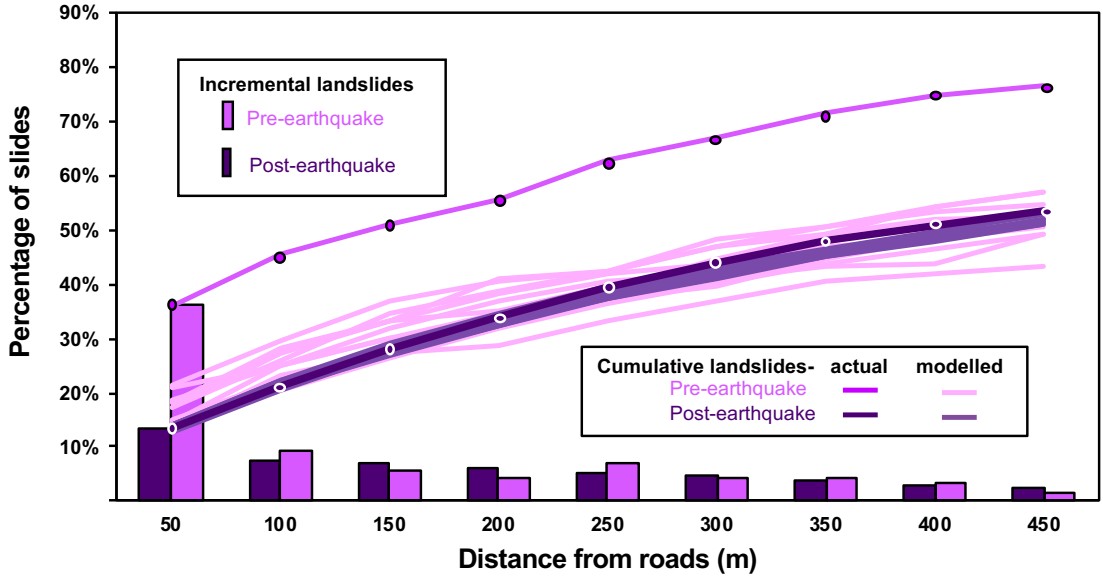

**Figure 5**. Distance from roads of earthquake, monsoon and randomly-generated landslides.  The red and blue bars
are the incremental percentage of pre- and post-earthquake landslides respectively that occur a given distance from
a road. The red and blue lines are the cumulative percentage of slides that occur at the given distances from the
road, and the light red and blue lines show the spread of the cumulative number of the modelled (n=10 runs),
randomly located landslides within the different buffer distances.
The shape of the curve that shows the cumulative number of landslides at increasing distances
from the roads in **Fig. 5** holds some additional information.   If there is a causative relationship
between roads and landslides, we might expect to see a change in slope of the cumulative
number of slides with increasing distances from the road that would correspond to a critical
distance where the mechanical influence of the road disturbance is reduced, and the number of
landslides begins to decrease (e.g. **Brown, 1987**).  However, we do not observe this change in
slope of the data, possibly due to resolution issues of the smaller slides.  The trend is not linear-
if we had a random distribution of roads across the landscape in addition to the randomly
distributed landslides, we would expect to see a linear increase in the cumulative number of
landslides with distance from the road.  What we notice instead is that there are fewer slides
further away from the roads than would be expected, suggesting that the roads might be in
locations that are predisposed to failure, such as near valley bottoms or ridge tops.
**4. Discussion**
Informal rural roads are causing dramatic changes in the physical and social landscapes of the
Middle Hills region of Nepal.  Although the number of slides generated by monsoon rains during
a given year is small when compared to the vast number of slides triggered by the Gorkha
earthquake, they nonetheless have a substantial impact on the physical and social landscape.
This study shows that there are twice as many landslides in the more developed areas (with its
good agricultural soils and vast network of informal roads) than there would presumably be if the
roads were better engineered.  The productive soils lead to more agriculture, and agriculture
benefits by having access to markets by way of roads.  As the population in this region will be
impacted by the proposed BRI trunk road, expansion of the informal, rural transportation
network is likely to follow, triggering more monsoon-rains driven failures, property loss,
transportation disruptions, and deaths.
The relationship between roads and landslides gives us an idea of how important these
anthropogenically-controlled slides are in shaping the landscape.  The risk of roadside failures is
heightened during the monsoonal rains because of slope oversteepening on the uphill side of
the road and the deposition of excavated debris on the downhill side that is easily mobilised
during heavy rainfall events (accentuated by runoff from the road- see **Sidle et al., 2006**).  To
make a stronger link to causation, it would be helpful to model how far the changes associated
with the road influence the failure mechanics. Regardless, this combined road-rainfall effect is
more acute than earthquake- generated failures in terms of percentage, if not total numbers.
These road-related failures also impact the sediment delivery system. While this snapshot of
monsoon-induced slides caused by informal roads is small compared to those generated by the
earthquake, it is important to consider this additional material in annual budget calculations
based on current river sediment load, and over longer periods of time.  There are many new
hydropower schemes following the BRI trunk road development, and they will be forced to
contend with this additional sediment burden.
China's BRI fits well with the Nepali government's long-term development strategy to promote
road development (**Murton, 2016; The Economist, 2017**).  While the roads constructed by the
Chinese in the Himalaya are well-engineered, informal and less well-engineered roads funded
by direct foreign investment and remittances have expanded significantly since the end of the
Maoist insurgency in 2006 (**MoF, 2016**). With the costs of rural roads managed by federally-
funded districts, scarce funds needed for road maintenance compete with the need for
investment in other sectors. **Leibundgut et al. (2016)** found that the economic impact of rural
roads around Phewa Lake, Kaski district of western Nepal amounted to $117,287 USD/year in
maintenance costs, forecasted to rise to $192,000 USD/year by 2030 with the current rate of
road construction.  Furthermore, over the last 30 years, tens to hundreds of deaths due to
landslides are recorded every year (**Petley et al. 2007; DesInventar, 2016**), and yet it remains
unclear how many of these failures are related to roads. Considerations of safer and more
sustainable "Green roads" that consider local engineering geology and best practices in design,
construction and maintenance (**Hearn and Shakya, 2017**) are outweighed by local communities
negotiating with limited funds, short-term political agendas and ease of access to heavy
equipment.
**5.  Conclusions**
The landslides generated by the 2015 Gorkha earthquake provide an opportunity to compare
the distribution of earthquake-triggered, 'natural' failures with those triggered by humans in a
landscape heavily modified by informal road construction.  By comparing earthquake-generated
failures and those caused by monsoonal rains before the earthquake with suites of randomly
located landslides, we show that there are likely to be twice as many monsoon-generated
landslides in terrain with poorly-constructed roads than would be present without roads. While
these anthropogenic slides do not represent a much of a change in the physical systems during
any given year, over time, their impact cannot be ignored. The socio-economic landscape,
however, is being severely impacted by an explosion of informal roads to the point where it is
hindering the socioeconomic development that the roads sought to bring and killing too many
people in the process.  Landslides in the Anthropocene are no longer simply a function of
seismology, geology, geomorphology and climate as poorly-built roads are rapidly changing the
landscape.
Better engineered roads will lead to more sustainable economic development, but these roads
come with a price.  Although foreign investment aids construction, maintenance costs fall on
impoverished communities who must decide between access and basic services.  Green
solutions such as plantings on metastable hillslopes are more economically sustainable and can
be implemented by community members with minimal training.  There is little that can be done
to control the tectonics or the climate, but economically feasible and environmentally sound
adaptations will reduce losses in resources and lives.

**Acknowledgements**
The authors would like to thank Yale-NUS College for supporting field research in Nepal.
Special thanks go to our colleagues A. Pang, S. Chee, A. Dominguez, and the students from the
Yale-NUS College Learning Across Boundaries Nepal programme.  Thanks also to C. van
Westen and M. Delalay for constructive ideas in the field, Z. Sandeva and K. Gurung for
logistics, J. Gruber for the modelling ideas, and R. Mukherjee (YNC) for geopolitical framing.

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
