# Peer review of "7091-Roads and landslides in Nepal: How development affects environmental risk"

_Natural Hazards and Earth System Sciences, 2017_

## Referee Comment (RC1) · Anonymous Referee #1 · 30 Jan 2018

This is a very short piece that looks at the spatial distribution of landslides triggered both by the 2015 Gorkha earthquake in Nepal and (presumably) by monsoon rainfall, and asks whether their locations are random or are correlated with distance from a road network. The links between landsliding and rural roads in Nepal have been addressed extensively in the past, both from physical and social science perspectives, and my first comment on the manuscript is that the authors need to clarify the novel contribution that they are making. The motivation for the manuscript is essentially a single clause in line 44 ('To better understand the link between geologic hazards and development, we examine...'), and I think that the manuscript would be greatly improved if they can explain, with a bit more care, why exactly they have done this and what specific niche or gap in understanding they are trying to fill. There is a good recent review of

road access and landsliding in the Himalayas that may help the authors to frame this: Hearn and Shakya (2017) Engineering challenges for sustainable road access in the Himalayas, Quarterly Journal of Engineering Geology and Hydrogeology 50, 69-80. I was also surprised that no mention was made of the Rural Access Programme, and the authors might want to have a look at a 2003 report on landslides and the RAP (http://rapnepal.com/report-publication/landslide-risk-assessment-rural-sector).

Addressing this concern need not add much text to the manuscript, but to me it should be done before this can be accepted for publication.

More specific comments and technical corrections, tied to line numbers in the text:

line 45: there's a word missing after 'landslides'

50-53: this sentence is key to understanding what the authors have focused on, but it is worded awkwardly – can they clarify what they are trying to achieve?

54: Roback et al. (2017 Geomorphology) mapped over 24,000 coseismic landslides – did the authors use the same data set? If not, is the landslide data set documented in Gnyawali and Adhikari 2017? Given that that latter reference appears to be in an edited book that some readers may not have access to, a bit more information on the coseismic landslide data set would be helpful. And it's a bit confusing to refer to landsliding as being controlled by 'rupture location' – what does that mean?

57-59: does this correlation imply causation? Let me put this another way: if the coseismic landslides were dominantly in bedrock, then why does it matter what soil type is found at the surface? Could this correlation be instead due to some other factor (e.g., steeper hillslope gradients associated with inner gorges along the major river valleys)? I think the authors need to clarify the proportion of their landslides that involve bedrock (as opposed to occurring within the regolith), or otherwise justify their use of this correlation in framing their work.

64: the authors could clarify that they are referring here to the magnitude-frequency

distribution, not the spatial distribution. And how many Monte Carlo simulations were done?

65: there's a potential problem here if the landslide areas don't match the size of the hillslope where they have been randomly placed, and if the hillslope sizes vary in some systematic (non-random way). It's hard to know whether this is an issue or not, because the reader isn't shown the distribution of landslide sizes.

66: it's not clear from the text here whether the buffers are incremental or cumulative. Fig 2 makes it seem as though they are cumulative, but that's never explained clearly in either the text or the caption. And the Openstreetmap data set for Sindhupalchok is quite detailed – which roads were included? Did the authors include footpaths as well or just 'engineered' roads?

70: there's a word missing after 'random'

73: I suggest cutting 'recent' as the time scale over which these landslides have occurred is not known (or at least is not given to the reader). There's no detail on how these landslides were mapped and the time period over which they occurred. Were they mapped from imagery or aerial photos, or some other method? Line 74 says that their 'area distribution' (= magnitude-frequency distribution?) is 'not too dissimilar' to that of the coseismic landslide data set, but that doesn't mean very much to the reader.

78: it's not clear from the text how both the coseismic and rainfall-triggered landslide data sets were used. Were both used to generate the fine lines on Fig 2? If so, which is which? I'd suggest differentiating them somehow – perhaps with solid and dashed lines – because the reader has not been shown the mag-freq distributions and so can't judge for themselves whether or not these really are drawn from similar distributions.

81: what do the authors mean by 'predictably'? Can they give an idea of what they'd expect? For example, if the landslide locations were truly random and the numbers in Fig 2 were cumulative, then you'd expect a linear increase in distance from the road.

Neither the rainfall or coseismic data sets seem to yield a relationship that is truly linear. I also wondered about weighting the data by landslide area, but that's probably a step too far for a brief communication.

86: Fig 1 lacks any coordinate grid, and panel A really needs some annotation (e.g., Bahrabise, Chaku, Kodari) to help orient the reader. The text 'to Kathmandu' and 'to China' can be cut – the latter especially because it is placed in China.

91: this text makes it sound like the buffers are incremental rather than cumulative, but that wouldn't fit with what the figure shows.

95: 'landslides'

107-109: there is another query here between correlation and causation. The authors have shown an apparently non-random association between rainfall-triggered (but not coseismic) landslides and road locations. To ascribe this solely to road construction, they would want to eliminate the possibility that roads are correlated with topography that is susceptible to landsliding. In other words, can they show more clearly that it's the presence of the road that is the causative factor?

131-133: this sentence is awkward and confusingly worded – can the authors clarify this (perhaps by separating the statistics)?

141-147: these conclusions have little to do with the analysis that the authors have shown, and introduce a few ideas (green solutions, 'environmentally sound interventions') that have not previously been discussed. I think it would be better to keep the conclusions focused on the novel contribution that the authors have made (thus linking back to their motivation in the introduction), and perhaps move coverage of green solutions to the discussion if it is relevant. Along those lines, I'd again ask them to consider the balance between bedrock and shallow landslides in their data sets – green solutions might be relevant for the latter but are unlikely to make a substantive difference to risk from the former.

200: the Petley et al. (2007) paper, while definitely relevant for what the authors are trying to do, isn't cited as far as I could see.

———————————————

---

## Referee Comment (RC2) · Anonymous Referee #2 · 2 Mar 2018

Summary This is a valuable piece of work that approaches the question of interactions between road building and landslides from a statistical perspective. This type of approach may be particularly useful in countries like Nepal where the physical processes may not be fully understood, due to issues such as informal road building and access to detailed data. I suggest that the paper should be expanded from a brief communication to allow more room to describe the data and methods so that others could attempt to repeat the analysis performed here, which would not be possible with the information provided at present. I have some concerns about the statistical approach used, which can most likely be addressed through a more detailed description which I outline below. Overall, I believe this paper will be a good contribution to the literature with appropriate medium level revisions.

[Figure]

Medium level comments:

Use of OpenStreetMap. Although I agree that OSM may be the most appropriate choice of data for an area where may of the roads are not 'official', I would like to know roughly how complete the authors feel OSM is for their area of interest. OSM can be highly heterogeneous in terms of spatial coverage and quality, so I would like to be assured there is no bias in the data (e.g., if one part of the study area has been mapped to great detail, and others have not, or roads have been traced reasonably accurately).

Landslide inventories used. In a statistical study like this, landslide inventory completeness will be key to fully understanding the spatial distribution of landslides. Please describe the methods used to create the inventories and give some indication of completeness.

Monte Carlo simulations. Please describe this process in more detail - e.g., how many iterations? Did you generate landslide areas, or just landslide point locations? If point locations, which part of the landslide do these correspond to?

Figure 2. I feel this figure should possibly be presented as a bar graph, or a clearer description is needed to explain the increments of buffer used. By using a continuous line to represent % of landslides at a given distance from the road, it implies that the total % of landslides adds up to more than 100%.

Random distribution of landslides. I am not convinced that landslides would be spatially randomly distributed within a given soil class. There is plenty of literature discussing how other factors such as topography control the distribution of landslides. I understand this is an assumption for the statistical model, but needs further discussion.

Distribution of roads. Similar to the above, I believe that the spatial distribution of roads will be linked to the landscape/topography. For example, are roads preferentially built in valley bottoms or ridges, or are they generally mid-slope? This may affect the number

of landslides that occur within a given distance of the roads. I would like to see some discussion of this in the paper.

Peak in landslide occurrence at 100 m from the road. I believe the finding that the distribution of landslides is different compared to a random distribution at around 100 - 200 m from the road is the most interesting finding from your work, but needs further discussion. What does it mean in terms of road building and physical process for landslide occurrence to peak at this distance from roads? This distance is considerably wider than the road plus the likely zone of influence either side (i.e., I would image that either side of a road, there would only be about 20 m maximum that is affected by road building).

Discussion and conclusion. Although interesting, the discussion and conclusion feel quite separate from the analysis, and do not particularly reflect on the findings. As mentioned above, a deeper discussion is required of why there might be a peak in landslide occurrence at a given distance. Following this, you can then go on to discuss this in light of road building policy.

Minor Level Comments:

I appreciate that it may be hard to find peer-review literature to support some of the statements made, but there are several places where there should be some citation. E.g., Line 46, 55, 67.

Line 43 'serves' should be corrected to 'services'

---

## Editor Comment (EC1) · F E Taylor (Editor) · 20 Mar 2018

Thank you to the authors for their interesting submission which investigates how the spatial distribution of landslides may be related to the spatial distribution of roads in Nepal. Thank you to both anonymous reviewers for their feedback on the paper. Both reviewers highlighted similar areas for improvement, many of which come down to having more space to explain the methods and data. For this reason, I suggest the paper format is switched to a full paper rather than a brief communication. With appropriate medium to major level revisions, this will allow:

- Further explanation of the data used for analysis (both the landslide inventory and road data) - Further explanation of the Monte Carlo technique used to model landslides

[Figure]

- Further discussion and justification of how the analysis reflects the actual distribution of roads and landslides in the region with respect to factors such as topography - Discussion of results reflecting the analysis that has been undertaken

In addition, both reviewers found that Figure 2 needed further explanation, and possibly some adjustment to better communicate the findings.

I believe the revised version of the paper will make a strong contribution to the special issue, and I look forward to reading this.

If you have any questions, please do not hesitate to contact me.

Regards,

Faith Taylor

---

## Author Comment (AC1) · 16 May 2018

I have responded to these comments in an earlier version, but respond again here as it was decided that the paper should be expanded from a Brief Communication to a full paper. Below are the updated comments.

Reviewer #1 (RC1) makes some very good comments and points out some extremely helpful papers that we were not aware of. RC2 suggests that the paper should be expanded past a Brief Communication- while I certainly agree, we feel that the timeliness of this paper with the coincidence of the special issue and the rapid expansion of roads with the Belt and Road Initiative (BRI) make it quite topical right now, and should get out

[Figure]

sooner rather than later with a more thorough treatment (which is indeed underway).

We have agreed to expand it to a full research paper.

RC1 points out that there have been links between roads and landslides have been addressed in the past- while this is indeed true, this paper seeks to point out the heightened need to pay attention to this well-documented phenomenon as the development increases a notch following the end of the civil war and the pressing forward of the BRI. As far as I am aware, this is the first study to compare landslides generated by an earthquake to landslides generated by monsoon rains through a lens of development.

RC1 points out a recent paper by Roback et al. (2018) that mapped coseismic landslides in this region. There are at least 4 different landslide catalogues that I am aware of, and we chose to use the one generated by our Nepali colleagues (Gynawali and Adhikari, 2017) that we have been partnering with since before the earthquake. We find it important to not only include our local colleagues in the paper writing process, but also use the data they collected as a way of building the necessary local capacity. Without entrusting our colleagues to do this critical work and see their work represented in the literature, and having only the work from large, well-funded researchers from overseas recognised, local scientists are actively disincentivised from doing their own research. While I would hope we could trust a published reference, we have described the methodology for identifying failures in the text.

On lines 57-59, RC1 asks about the correlation between landslide location and soil type, and specifically mentions the possibility of bedrock failures. We expanded this in the text, noting that the vast majority of slides that we groundtruthed in the field involved only the regolith, we were not able to assess all slides, and our sampling would have been biased as we were traveling along roads. It is exceedingly difficult to identify bedrock failures from satellite data alone. Furthermore, while RC1 is correct that this is a strong correlation, being critical of that fact does not rule out causation. As there will be more landslides (shallow) where there are soils in steep terrain, the soil is weaker

than bedrock (which because of climate hasn't developed soils. . .), therefore is more likely to fail under the same stressors. This is an observation based on the data as shown in Fig. 2. The distinction of soil type and land use is significant, since nearly all economic activity in this region is agriculture-based, therefore that is where most of the risk is located. Because the sample size of the earthquake (EQ) landslides is large compared to the pre-EQ slides, there are quite a few EQ slides in unpopulated areas, and those might skew the results. As those are not pertinent to the question of development and risk, we cut them out based on the fact that development occurs where there are resources available to support it.

RC1 states that there may be a problem if the landslide areas don't match the size of the hillslope where they have been randomly placed. I think that they are referring to the possibility that a landslide could be large enough to exceed the size of the hillslope it occurs on (but I am not entirely clear about that). Most slides are quite small as compared to the scale of the topography of the region- we have sought to clarify this by adding an inset to Figure 2 that shows the histogram of the slide areas. We also noted that we did 10 Monte Carlo simulations for each class of slides (EQ vs. monsoon triggered- thanks for catching that).

---

## Author Comment (AC2) · 16 May 2018

While we have responded to these comments in the earlier version of this submission, it was decided that the updated paper should be a full submission rather than a Brief Communication. The comments below reflect both the initial submission and the updated version.

Both RC1 and RC2 raised questions about the Open Street Map data. In response to RC2, the OSM data is remarkably comprehensive- we ground trothed it using both recent Google Earth imagery as well as field observations. For RC1's question about are they footpaths or bulldozed roads- this is an excellent question, and all we can state is that they are all roads/path large enough to be seen on satellite, hence we assume

[Figure]

will have a more significant effect on the physical landscape than smaller, less well-travelled paths. Our Nepali colleagues have since confirmed that these are the roads that are capable of supporting some kind of vehicle. (Now, anyone who has been to Nepal is right to question this!)

RC2 questions the 100 m distance as being considerably wider than the road plus the likely zone of influence on each side. We think it is a bit beyond the scope of this short paper to delve into the details of specific slides, but we can assure RC2 that the zone of landsliding can extend hundreds of meters past the road itself- runout zones of debris mobilised during rainfall failures can be kilometres, and retrogressive failures can extend far upslopes. In fact, we were quite surprised to see how more of the EQ landslides didn't intersect roads!

We were hoping to see that magical peak of landslide occurrence at a certain distance from a road, however the data were not cooperative. As this paper is an attempt to look at the risk of landslides to exposed communities, not just understanding the physical hazard that the landslides represent, we feel it necessary to frame the results in the context of how attempts to make communities less vulnerable (economically, socially and physically) by constructing roads is actually making them more vulnerable as far as exposure to landslides hazard is concerned. We will leave it to the social scientists to sort out if the relative gains achieved by constructing roads (along with the associated landslides) exceed the losses of property and life that comes every monsoon season. Therefore, we would like to keep the analysis, recognising a shift from hazard focus to a more nuanced and complex treatment of the associated risk.

We agree with the reviewers and the editor that the conclusions did not match the majority of the paper. We have tried to rectify this with more analysis of the data presented while sticking with our goal to highlight how the hazard and the risk to communities are tied together. Here, communities are affecting the hazard, and vice-versa.

2017-461, 2018.

---

## Author Comment (AC3) · 16 May 2018

Response to Editor

We have agreed with the editor that this paper should be submitted as a full paper rather than a brief communication. As such, the paper is largely rewritten, and many of the comments are addressed in the expanded version. Below, we will endeavour to address the comments, but perhaps less with specifics (line-by-line) and more with the big picture.

Point 1- Further explanation of data and Monte Carlo. We have added a more detailed description of each. The description of the landslides was done by our Nepali co-author (KRG) using a method that was very similar to that described in detail by Roback et al.

[Figure]

As we have not seen the Monte Carlo method that we used in this paper described in the literature, we have tried to describe it in sufficient detail for it to be repeatable, but not so much detail so that it drowns the results.

Point 2- The analysis of the results is complicated by the fact that this is not a pure hazards paper. Instead, we seek to see how the hazards affect people and, critically, vice-versa. In this particular case, it is not possible to divorce the two. The analysis we seek to complete is not so much about understanding the mechanics of the hazard, but rather the effect the increased hazards have on exposed communities. As such, the distribution with topography (did it occur on a steep slope? Was it at the bottom or top of the slope?) is less important than the fact that wherever the communities are that built these roads are located, they still built the roads and are therefore by their sheer existence, tied to the associated hazards. If this were a paper that focused on the distribution of slides, amount of additional sediment delivered to streams, etc., we would agree that the relationship with topography would be key to predictability.

Other, big picture changes. With more space, you will notice that we decided to add a figure at the beginning (Fig. 1) that shows what this actually looks like in the field, along with a schematic that shows the different modes of failure, and a couple examples of two of those modes. As far as we are aware, this simple schematic has not been done previously.

We have cleaned up Fig. 2 with landmarks (towns, river names) based on suggestions by the reviewers, and made the inset (Fig. 2b) slightly less psychedelic looking (purple to grey).

The new Fig. 3 now shows both the cumulative and incremental number of slides at given distances from the road. This has the benefit of clearly showing that the main driver of the discrepancy is the number of monsoon-triggered slides that are within 50 m of a road, clearly demonstrating the genetic relationship.

We have added a Fig. 4 that shows the relationship between road construction, the

increase in Foreign Direct Investment following the end of the Maoist insurrection, and associated landslide deaths. This shows how if the trend of increased road construction is magnified by the Belt and Road Initiative, the upward trend of landslide deaths is also likely to increase.

—————————————————————

---

## Referee Report (RR1)

Review of NHESS-2017-461 McAdoo *et al.* Roads and Landslides in Nepal V2

**Overview:**

This paper presents interesting analysis of the interaction between informal road building and landslide occurrence in Nepal from a statistical perspective. There is little in the literature using methods like this, and I believe there is merit in the approach, so there would be real value in bringing this into the literature in a scientifically evidenced way. The paper has changed considerably since the first submission, with addition of figures and longer text. Because of the considerable difference between the first and second submissions, I have reviewed the paper thoroughly as if it were a first submission. I am keen to see the paper published, and understand the authors' wish to contribute to the debate on this timely topic. However, this cannot come at the cost of scientific rigour and use of evidence, which needs further work. I support the authors' comment that this is a collaborative piece of work with Nepali colleagues where issues of capacity and incentive to publish are a challenge. However, I believe this makes it all the more pertinent to publish a scientifically rigorous and evidenced piece of research that becomes widely cited.

The paper has three key areas for improvement which I outline below, followed by a list of minor revisions which link into these key points. In the previous response to reviewers, each comment was not addressed individually, which made it hard to follow the link between each comment and the action taken. I would appreciate it if each of my comments could be responded to individually, including my original comment in text. To aid this, I have numbered my comments. In the next revised version, I would appreciate it if the authors could use line numbering, as it has made it difficult to indicate precisely where I am commenting.

**General comments:**

*AR1: Being specific and evidencing.* The style of writing is somewhat informal and conversational. Rather than saying 'thousands of landslides', stating that your results are 'compelling' or saying a concept is 'well known', be specific, give exact numbers and let the results speak for themselves. There are numerous statements made with no reference to the literature, and only 11 peer-review pieces of research in the reference list. There are several places in the text where the implication is that the reader should go away and read another paper to understand what has been done here. Particularly for a readership from the Global South, it may not be possible to access these papers, and more generally makes it difficult to replicate the results. The authors must demonstrate that their research builds upon a body of evidence by citing existing literature and making closer links to this literature.

*AR2: The landslide inventory used.* The authors state a preference to use an inventory of landslides created by Nepali colleagues that is within the peer review literature, which is understandable. However, publication of an inventory is not the only criteria to ensure an inventory is suitable for the type of analysis performed on it. See Guzzetti *et al.* (2012) for a review of issues around landslide inventory production. The authors must give more detail on the inventory production methods to indicate that this is statistically robust for the type of analysis they are doing. Other highly cited papers performing statistical analysis of landslide inventories (e.g., Stark and Hovius, 2001; Malamud *et al.*, 2004) describe in considerable detail the previously published inventories they use. The item that concerns me most about the inventory you use is that there is no indication of time scale given for the landslides mapped pre-earthquake. Smaller area landslides tend to be erased from the landscape more quickly than larger area landslides, so it is difficult to make interpretations about the area and spatial distribution of these landslides without assurance that this is a substantially complete inventory.

*AR3: Replicability of methods.* The Monte Carlo simulation method is interesting, but not all of the NHESS readership will be familiar with this methodology. Indeed, I am familiar with Monte Carlo methods but could not replicate your analysis without further information about how you generated the random landslides. Neither am I convinced that a completely random distribution of landslides is appropriate for comparison, as real-world landslide locations are conditioned by many factors. There appears to have been no consideration of the number of Monte Carlo simulations required to be statistically robust. Overall, it remains difficult for me to fully comment on the methodology as it is still somewhat vague.

**Specific comments:**

*AR4: Road data.* Throughout the paper, it is not entirely clear how you make the distinction between formal and informal roads in a systematic way. At times, it is implicit that informal roads are funded by foreign investment. I suggest in the introduction to more explicitly explain what these informal roads are and how you distinguish them. I also see that in

the response to reviewers, you have stated that you are confident that the OpenStreetMap data for this region is robust and you performed some testing of this. If this is the case, state it in the paper and give more detail.

*AR5: Figure 1*
- Are these images all the authors' own?
- "Deeper seated landslides that are accommodates by" typo.
- Each figure should be labelled 'A', 'B', 'C', 'D' and then the top right figure relabelled I, II, III or something similar. This makes it easier to refer to the figure in text and easier for the reader to quickly distinguish between descriptions.
- Top left figure. This is an interesting diagram. This should be made into a separate figure and enlarged. It also needs to be clearer what the basis for this figure is in terms of evidence. Give some indication why this is a complete set of possible landslide types near informal roads.

*AR6: Paragraph starting 'Many villages in the Middle Hills region'.* First sentence needs some supporting evidence.

*AR7: DesInventar.* Although DesInventar is a useful source of data, it has methodological biases which are key when interpreting results. I would suggest briefly describing that DesInventar records primarily come from media sources, so may be biased towards particular locations. I would also suggest giving slightly more detail for the in-text citation, e.g., DesInventar Nepal Database.

*AR8 Method paragraph 1*
- Without some indication of timescale and the inventory used, it is not convincing to state that landslides mapped pre the Ghorka earthquake are all triggered by the Monsoon. The region was seismically active before 2015, and there are other drivers of landslides aside from over-steepened roads. I suggest introducing the inventory and then discussing what is implicit about it. It may be useful to draw out additional key points from Petley (2007) rather than simply to direct the reader here.
- State why "Landslides generated by the earthquake…respond more to the geomorphology of the landscape" – the mechanism for this is not clear.

*AR9: Methods paragraph 2:*
- Not entirely clear what you mean by 'discrete' landslides, and how this differs to the landslides triggered by the earthquake
- "The landslide inventories we used was created" – typo – were created?
- Generally more needs to be said about this inventory. Creating an inventory of triggered landslides is not a trivial task. How do we know this is substantially complete? This is important for looking at spatial patterns. One option would be to look at the frequency size statistics of landslide areas and compare to already established distributions.
- Be more specific – how many is 'many landslides'? What percentage is the 'vast majority'
- For mapping landslides pre-earthquake, did you use just one image? If so, what is the date of this image? Can you estimate how many of these were relatively fresh (i.e., seasonal) versus older landslides?
- Landslides (particularly smaller ones) tend to be erased from the landscape over time so I am not convinced about using this 'pre-earthquake' inventory to look at the spatial distribution of landslides, as this may be incomplete. If it is a case that the pre-earthquake inventory is primarily from one season, this might be more reasonable, but needs to be explained.

*AR10: Methods paragraph 3:*
- The distribution of landslides – state whether you mean spatial, statistical or other type of distribution
- State how you compared the distribution of landslides in your inventory to other earthquake triggered inventories.
- It would be useful for this paper to have a specific section or table on data – state what the sources of all the data are that you are using – e.g., soil types, and give more information about the data you have created (the landslide inventory).
- I see the reasoning that there is more agricultural development in the productive soils, but this needs to be backed up by evidence. There are other data products (e.g., croplands.org) you could use to estimate

agricultural or built-up areas to make this statement more robust. Nepal is also rapidly urbanising, which is a different process that may result in road building around small towns and large villages. I believe this should be considered in addition to agricultural areas as an indicator of human impact on the landscape.

*AR11: Methods Paragraph 4:*
- Be cautious about stating that the correlation between landslide and road occurrence suggests causality.
- I am not convinced about the comparison of landslides in proximity to the road to a random distribution. Landslide location is conditioned by many factors and I do not believe landslides occur randomly across the landscape, even controlling for the location of roads. As a minimum, this needs further explanation and justification in the paper.
- State the method used to measure whether observed landslides match the randomly generated ones. This is not clear, as the location of randomly generated landslides will be different on each Monte Carlo iteration. State what part of the landslide you measure (centroid? Crown? Toe?) in relation to what (distance to nearest road, count within a buffer of a road?).
- Sentence starting 'the pre-earthquake landslides have a normal distribution' – distribution of what? Size, distance?
- State the resolution of the Google Earth imagery (which is typically sub-metre). I would be surprised if you are missing all but a small portion of the smallest earthquake triggered landslides due to issues of resolution, and believe this may be a result of removal of smaller landslides from the landscape by erosion, revegetation, ploughing etc.
- State how you have performed the curve-fitting of these distributions and tested the goodness of fit.
- Why only 10 runs of the Monte Carlo simulation? See e.g., http://kb.palisade.com/index.php?pg=kb.page&id=125 for a discussion of number of iterations versus confidence intervals.

*AR12: Results paragraph 1:*
- You did not mention the fieldwork in the methods section, it seems slightly odd to start the results by discussing field observations.
- The second (long) sentence of this paragraph needs splitting and expanding upon. It was not clear from the methods that you focus the analysis on agricultural areas and why this is done.
- What is a 'genetic' relationship?

*AR13: Results paragraph 2:*
- Why state a range (20-25%) for earthquake triggered landslides occurring within 100m of a road, but no range for pre-earthquake triggered landslides? Be specific rather than stating 'nearly 50%'.
- Not clear why you are discussing the total area of landslides, particularly when the sample size is two orders of magnitude different between the monsoon and earthquake triggered landslides.
- As stated previously, I believe the difference in average area may be due to the difference between analysing a triggered versus multi-temporal landslide inventory. For distributions that span multiple orders of magnitude and are skewed, it may be more appropriate to analyse the mode or median landslide area.

*AR14: Figure 3*
- Add legend indicating difference between bars and lines
- Add description of sub-figure in the figure caption
- Ensure axes are appropriately labelled in sub-figure. Unclear what has been normalised and why.

*AR15: Results paragraph 3*
- This paragraph is very conversational in style and needs tightening up, e.g., what is a 'kink in the trend'. Why 'borrow from the fractal literature'?
- This discussion about a 'crossover length' is unclear – explain in the text why one would expect to see a decline in number of landslides at a given distance from the road.
- Generally, the concepts in this paragraph are interesting but need further explanation and possible supporting evidence.

*AR16: Results paragraph 4.* Give evidence to support the statement that the roads follow river valleys and ravines

*AR17: Discussion paragraph 1*
- In the methods, state how you systematically separated out informal and formal roads from the OSM dataset
- In the results section, I suggest presenting a brief quantitative analysis of kilometres of road length per soil type to support the statement 'with its good agricultural soils and vast network of informal roads')
- As per comment AR10, in addition to agriculture, urbanisation is another form of development. At present, you imply that good soil is the only control over which areas are developing (and where there are more roads). I believe the language needs adjusting (or analysis also performed on small towns and large villages) to reflect that soil quality is not the only indicator of development.

*AR18: Discussion paragraph 2.* If something 'is well known', then add citations to support it. More broadly, there are very few references in the discussion to frame your results in terms of previous work done on this topic.

*AR19: Discussion paragraph 3.* As per AR9 and AR11, the size (or area) of landslides is an area where a lot of work has been done (e.g., Stark and Hovius, 2001; Malamud *et al.*, 2004; Stark and Guzzetti, 2009 amongst others). I believe it is possible that your findings in terms of landslide size may be a result of sampling rather than process necessarily.

*AR20: Discussion paragraph 4*
- This is a little confusing to introduce the Maoist insurgency here without any prior discussion of the insurgency or its implications for road building.
- Generally in this paragraph, the discussion about correlation between deaths from landslides and increase in road length sounds more like results than discussion, and would benefit from evidencing.
- I am concerned about the use of DesInventar here to imply an indirect link between political regime and deaths from landslides. As mentioned previously, much of the data for DesInventar comes from media reports, and thus has biases. The links between politics and journalism are of course too complex to discuss in detail in this paper, but there needs to be some acknowledgement from earlier on in the paper that DesInventar has biases and this uncertainty acknowledged when discussing DesInventar. I suggest that Aryal (2012) should be read and possibly cited to give some context.

*AR21: Figure 4*
- Give legend entries titles
- In legend, state deaths from landslides (otherwise it implies total deaths from all causes)

*AR22: Conclusions*
- Without further discussion in previous parts of the paper, I am not convinced that you are comparing datasets of human versus natural triggered landslides
- Some of the conclusions are introducing new ideas and read more like a discussion

I would like to reiterate that I genuinely found the approach of the paper novel and interesting, which is why I have put a lot of effort into a complete review. My comments primarily relate to better communicating the research and demonstrating that this work has not been done in a vacuum. I look forward to seeing the revised paper.

**References Cited:**

Aryal, K.R., 2012. The history of disaster incidents and impacts in Nepal 1900–2005. *International Journal of Disaster Risk Science*, *3*(3), pp.147-154.

Guzzetti, F., Mondini, A.C., Cardinali, M., Fiorucci, F., Santangelo, M. and Chang, K.T., 2012. Landslide inventory maps: New tools for an old problem. *Earth-Science Reviews*, *112*(1-2), pp.42-66.

Malamud, B.D., Turcotte, D.L., Guzzetti, F. and Reichenbach, P., 2004. Landslide inventories and their statistical properties. *Earth Surface Processes and Landforms*, *29*(6), pp.687-711.

Stark, C.P. and Guzzetti, F., 2009. Landslide rupture and the probability distribution of mobilized debris volumes. *Journal of Geophysical Research: Earth Surface*, *114*(F2).

---

## Author Response (AR2)

McAdoo et al., "Landslides and Development", Response to Review

This is perhaps the most comprehensive, constructively critical review I (BGM) have received in 25+ years of paper writing. Thank you.

We have attempted to address all of your concerns, even if we respectfully disagree with a minority. We have included both the "track changes" version under the "supplement" based on AR5's comments, however subsequent minor additions and fixes based on the co-authors' comments have been wrapped into the final, untracked document (it just got way too messy after the major revisions).

The core concern of this paper is that it makes a link between causation based on a very strong correlation of rainfall-triggered landslides and poorly engineered roads. The literature is surprisingly weak on making this jump, and a geoengineering treatment of this relationship (that is simply accepted in the other papers citied in our study) is beyond the scope of this work. We hope that the findings of this study are convincing enough to justify our novel methodology.

Below we go through AR5's comments one-by-one, describing the changes made in the manuscript. We firmly believe that this is a much-improved manuscript based on the reviewer's comments and hope that we have sufficiently addressed each and every one. If there are any questions or concerns, please communicate them to me, writing on behalf of my co-authors.

Sincerely,

Brian G. McAdoo

~~~

**AR5: Figure 1**

Are these images all the authors' own?

- "Deeper seated landslides that are accommodates by" typo. Fixed.
- Each figure should be labelled 'A', 'B', 'C', 'D' and then the top right figure relabelled I, II, III or something similar. This makes it easier to refer to the figure in text and easier for the reader to quickly distinguish between descriptions. Agreed. Fixed.
- Top left figure. This is an interesting diagram. This should be made into a separate figure and enlarged. It also needs to be clearer what the basis for this figure is in terms of evidence. Give some indication why this is a complete set of possible landslide types near informal roads. Based on the reference (Sidle et al., 2006) and the combined experience of the authors, this is what we have observed in the field. As a relative newcomer to this field (BGM), I was surprised that I could not find a schematic that outlines the modes of failure associated with rural roads. We would all be keen to know if there is a better reference for this. However, as this is not a major focus of the paper, it is helpful for the reader to see what kind of failures are being considered, yet we feel it is better in the context of the examples in Fig. 1.

AR6: Paragraph starting 'Many villages in the Middle Hills region'. First sentence needs some supporting evidence. I am trying to read into this simple request, however I am challenged to find a way to prove that the villages connected by footpaths (instead of roads?) are indeed connected by footpaths- they just are. And I also think that it is quite logical that footpaths would be more socially and economically limiting than vehicular roads.

*AR7: DesInventar.* Although DesInventar is a useful source of data, it has methodological biases which are key when interpreting results. I would suggest briefly describing that DesInventar records primarily come from media sources, so may be biased towards particular locations. I would also suggest giving slightly more detail for the in-text citation, e.g., DesInventar Nepal Database. Agreed. However, as is the case in every loss database that we are aware of, data quality is questionable at best, hence the vague language ("scores" of deaths) and no mention of location. To address this concern, we have added a footnote (if this is acceptable to the editors) so as not to complicate the parenthetical citation.

**AR8 Method paragraph 1**

- Without some indication of timescale and the inventory used, it is not convincing to state that landslides mapped pre the Gorkha earthquake are all triggered by the Monsoon. The region was seismically active before 2015, and there are other drivers of landslides aside from over-steepened roads. I suggest introducing the inventory and then discussing what is implicit about it. It may be useful to draw out additional key points from Petley (2007) rather than simply to direct the reader here. Agreed. I have removed the (confusing, even to us!) interpretation about the roads causing these landslides. However, the overwhelming majority of landslides occur in the Himalaya during the rainy season. This is known, yet we reference Petley et al. (2007) for support. The majority of landslides generated by previous earthquakes would have been long-since covered by vegetation or development in this monsoonal climate.
- State why "Landslides generated by the earthquake...respond more to the geomorphology of the landscape" the mechanism for this is not clear. Agreed. Reviewing both Roback et al and Gnyawali and Adhikari, I think this statement was incorrect as previously written.

AR9: Methods paragraph 2:

- Not entirely clear what you mean by 'discrete' landslides, and how this differs to the landslides triggered by the earthquake Agreed. Removed "discrete".
- "The landslide inventories we used was created" typo were created? Agreed. Fixed, "inventory" to match the verb.
- Generally more needs to be said about this inventory. Creating an inventory of triggered landslides is not a trivial task. How do we know this is substantially complete? This is important for looking at spatial patterns. One option would be to look at the frequency size statistics of landslide areas and compare to already established distributions. Agreed. We have compared our landslide statistics with the Roback paper- While we seem to pick up more smaller landslides, these are not significant in the interpretation of

the results. We have added the figure below to the text, along with a modified beginning of the methods paragraph 2.

- Be more specific how many is 'many landslides'? What percentage is the 'vast majority' This is quite difficult as we do not have a count of the ground truthed landslides. Our ground-truthing was based on a rapid visual/spatial identification rather than a comprehensive analysis of each landslide. During field trips up the Arniko Highway/ Bhote Koshi river valley, we had maps and GPS in hand, and were able to visually identify the nearby roadside landslides as well as the larger failures across the valley. I will note that (surprisingly) the other remote sensing papers (Roback et al., Martha, et al.) did not go into detail with the specific mode of landslide failure as these were both based entirely on remotely sensed data. Our paper is also based on satellite data, but we have the advantage of being based in the region and hence these sites are more easily accessed.
- For mapping landslides pre-earthquake, did you use just one image? If so, what is the date of this image? Can you estimate how many of these were relatively fresh (i.e., seasonal) versus older landslides? Multiple images are used to create the landslide inventory. As we are concerned with a binary temporal distribution (pre- vs. post-earthquake) versus a finer grained study of the pre-earthquake landslides, we hope it isn't necessary at this point to document the specific dates of the imagery that corresponds to the mapped slides- This too is quite difficult in Google Earth as different regions within the area of interest may be covered by different images from the same general time period. An interesting follow up study that uses this methodology might consider mapping all the landslides visible prior to the earthquake over time. However, that is beyond the scope of this study.
- Landslides (particularly smaller ones) tend to be erased from the landscape over time so
  I am not convinced about using this 'pre-earthquake' inventory to look at the spatial
  distribution of landslides, as this may be incomplete. If it is a case that the preearthquake inventory is primarily from one season, this might be more reasonable, but
  needs to be explained. We have clarified the methods we employed to ensure that we

have, 'caught' the smaller, older landslides. As this study is not concerned with the temporal distribution (pre-EQ), it is less critical that we catch all of the older slides.

**AR10: Methods paragraph 3:**

- The distribution of landslides state whether you mean spatial, statistical or other type of distribution Area and spatial (geographical). This is clarified in the text.
- State how you compared the distribution of landslides in your inventory to other earthquake triggered inventories. Based on your helpful suggestion, we created a histogram of the landslides in our database to compare it to that in the Roback dataset. While the tail ends match remarkably well, the smaller slides diverge a bit, suggesting we picked up more smaller slides than they have. See new Figure 2.
- It would be useful for this paper to have a specific section or table on data state what the sources of all the data are that you are using – e.g., soil types, and give more information about the data you have created (the landslide inventory). We agree this would be helpful, but as the sources of data are identified in the references, we are not clear what added benefit a table would add. The post-earthquake landslide database is from Gnyawali and Adhikari, the pre-EQ database is unpublished, but described in this study, and the soils are from Dijkshoorn and Huting. We hope that the additional description of the pre-EQ landslide database is sufficient for the reviewers/editors.
- I see the reasoning that there is more agricultural development in the productive soils, but this needs to be backed up by evidence. There are other data products (e.g., croplands.org) you could use to estimate agricultural or built-up areas to make this statement more robust. Nepal is also rapidly urbanising, which is a different process that may result in road building around small towns and large villages. I believe this should be considered in addition to agricultural areas as an indicator of human impact on the landscape. The evidence for more agricultural development in the productive soils is a simple correlation, and we don't feel it is too much of a jump to imply causation. Dijkshoorn and Huting point out that these units are agriculturally productive based on the FAO classification, and we see from a GIS overlay that these soils correspond to terracing, villages, roads, etc. In this area of Nepal, the primary industry is farming. As larger villages develop, their primary purpose is to provide services for the surrounding industry (farming). Unfortunately, the croplands.org data for Nepal is wanting- vast areas of rice terraces in the Middle Hills are missing while 'cropland' shows up at 4200 m on the very arid Nepal-China border.

---

## Author Response (AR3)

Dear NHESS Editors,

Please find attached a point-by-point response to how we addressed the editor's/reviewer's comments. Of course, let us know if you find that any of the points below are insufficiently addressed.

Thank you,

Brian G. McAdoo
(for the authors)

**Suggestions for revision or reasons for rejection (will be published if the paper is accepted for final publication)**

As per my previous comment, I believe Figure 1b should be a standalone figure and needs a legend to indicate the root zone. I am not aware of any similar figures, hence suggesting this should be standalone. It would make the paper more citeable if others can use this figure to describe the processes of landsliding near to roads.

Fixed. We have also included a note that the second mode might also include water seepage into fractures and joints that could facilitate deeper bedrock failures via freeze-thaw (IIb).

Since writing your paper, Froude and Petley have published a new paper on the topic of landslide fatalities. As you cite Petley et al. (2007) It might be worth referring to this updated paper as it supports your findings.

Updated.

Line 70. This statement that there has not been a study of landslides and roads in Nepal since 1979 is too definitive. For example, Bhandary et al. (2013), Wagner et al. (1988) and others seem relevant. I suggest changing the text to something like 'there have been a limited number of studies..'.

We have added Bhandary et al., however we could not locate Wagner et al. (1988). There is an unpublished Wagner et al (1983) cited in Bhandary's paper, however as it is unpublished (and quite difficult to locate), so we will leave it with Bhandary.

Line 83, the term 'scores' may not translate well. Change to 'tens'.

Done.

Line 94 'we present compelling evidence'. I would remove the word compelling as it is not particularly neutral or scientific.

Agreed. "Compelling" deleted.

Sentence starting on line 109 needs rewording. Roback et al do not investigate differences in distribution of earthquake and rainfall triggered landslides or the influence of human factors. They state that natural drivers appear to influence the distribution of earthquake triggered landslides, but there is no comparison to rainfall triggered landslides, so this is all you should state there.

We have changed the wording as requested, but highlighted the lack of consideration of the human alteration in Roback to help set up the next sentence.

Paragraph starting line 135. Give some indication of the area (or better still, normalise by area) of each of these soil types.

We have added some normalised values for comparison.  Although this makes the text more busy, and it might benefit from a separate table, the numbers add a clarity to why we limited our analysis to these soil types.

There are two figures named figure 2.

Fixed.

Line 103. In this paragraph, please briefly state what you constitute to be landslide area and ensure this is consistent for both inventories (e.g., individual landslide polygons containing both source area and run out, or just source area?).

We have added "and deposits where visible" after "scars" in the text.  This is indeed the case in both inventories.

Line 170. What is the log-normal distribution of landslide areas? I see this in the author response but not in the paper before this point, so it feels a little 'out of the blue'. Landslide areas more typically have been shown to follow a Double Pareto or Inverse Gamma distribution. State what distribution you used for your post-earthquake landslides.

We chose a simple log-normal distribution based on the fit of the observed areas vs. the model, which gave us an $R^2$=0.96 and 0.94.  Therefore, we have added the $R^2$ value to the text to justify the choice in curve fit.

Line 194 the number of landslides is small by comparison... to the earthquake triggered inventory.

Fixed

Line 194 rather than saying landslides cover a total area, state that the total area of landslides is x. Otherwise this gets confusing about the overall area in which the landslides occurred.

Fixed

Line 196. Rather than the average, state the median or mode. Landslide areas span many orders of magnitude and are non-normally or symmetrically distributed so the average is not particularly useful.

Agreed, and I chose to leave in the average to highlight your point about the asymmetry.

Line 196. The average area of the rainfall triggered landslides will most likely be higher because this is not a triggered event inventory and includes some older landslides rather than there being any physical reason behind this. Smaller landslides tend to be eroded away more quickly, leaving only the largest area landslides, which affects the distribution. If you plot the probability density distribution of the rainfall triggered landslide areas, you will see that it is most likely shifted towards the right-hand size of the axis (towards larger landslide areas). See Malamud et al. (2004) for further explanation. **The simplest solution would be to remove reference to landslide areas here as your results about roads are more interesting.**

We do want to highlight the correlation (and arguably causation) of the soil types and landslide distribution, so we left the total area numbers in there, reminding the reader that the smaller landslides may have been covered (Malamud et al., 2004).

Figure 4A belongs with Figure 2 (probability density plots) and would be best presented in the same format, which aids in visualising the broad range of landslide areas which span several orders of magnitude. Figure 4B then becomes separate. These figures are not particularly related, so do not belong together.

Done.

Figure 4B incremental percentage lines need point makers at each distance to indicate that this is not a continuous distribution. As I understand it, this line is showing the cumulative percentage of landslides within buffer zones, not the raw distances from each landslide to the nearest road.

Done.

Lie 259. Remove reference to average landslide sizes. This is most likely a result of sampling rather than process.

Agreed.  Removed.

Reference list: Please check all references are complete and in the appropriate format.

[revised manuscript text omitted]

---

## Author Response (AR4)

Dear Dr. Taylor,

Please see the response to editors as written by me on behalf of my co-authors. I have inserted responses to your comments below in purple text for clarity.

Thank you for the considerable time and care that the editors and reviewers have put into making this a much better contribution.

Sincerely,

Brian G. McAdoo (for the authors)

Editor Decision: Publish subject to minor revisions (review by editor) (26 Oct 2018) by Faith Taylor Comments to the Author: Dear Brian and co-authors,

Thank you for your revised version of the manuscript NHESS-2017-461. I am satisfied that you have implemented the majority of the reviewer comments and am pleased to accept the paper to publication subject to a small number of revisions. Revisions (a)-(e) below are small, technical revisions. However, I would like to see your revision with regard to comment (f) below before accepting for final publication.

(a) Please check the figure numbering throughout - both for the figures and reference to the figures in-text. At present you have 2 x Figure 2 and refer to a Figure 3b which is not present.

Figure and references double checked- please see 'trackchange' document for changes.

We also decided to switch figures 3 and 4 (in the new numbering scheme)- figure 3 is an overview of the mapped landslides, and figure 4 shows the histograms of those landslides so the logic just flows better.

(b) Line 123, sentence starting 'the landslide inventory we used'. Please state which inventory you are referring to (I think it is the pre-earthquake inventory, or perhaps it is both inventories, in which case say this).

**Fixed. Good catch.**

(c) Line 126 sentence starting 'the post-earthquake landslide inventory', please add the appropriate reference at the end of this sentence - otherwise in the following figure 2a (actually figure 3a), it is not clear which of these inventories is the one you actually use.

Adding, "pre- and post-earthquake landslide inventories" in the previous sentence clarifies which inventory we are discussing here.

**Figure reference fixed.**

(d) Line 127, sentence 'consists of scars observed'. I think this should be 'scars and deposits' for consistency with the previous statement.

**Fixed.**

(e) What is currently labelled as Figure 2b. This is not currently referred to in-text (unless this is an issue with figure numbering). Either refer to in text, or possibly remove. I am reading in printed greyscale and it is not possible to distinguish between pre-and post-inventories, so if you are editing the image, I would appreciate you adding texture/stronger colour contrast to Figure 2b.

Figures have been redone with a colour scheme that is more visible when printed in black and white. Because there are several overlapping colours in the map in Figure 3, we chose a purple that contrasts with the green of the RGe soil type, and we wanted to maintain consistent colouring of the pre- and post-earthquakes landslides in Figures 4 and 5. While I find the purple somewhat less aesthetically pleasing, they are indeed more visible when viewed in black and white.

(f) Sentence starting on line 181. As per previous reviewer comments, the description of the log-normal distribution remains unclear. Please add a sentence explaining how you chose the log-normal distribution. To replicate your results, readers need to know the parameter values used in the log-normal distribution.

We agree with the reviewer and hence have decided that more than a sentence is necessary to make it truly repeatable. We include the equations used, along with the residual standard error for both datasets, and explained the method in more detail.

I look forward to receiving the revised manuscript.

[revised manuscript text omitted]